# The Ceramide-Dependent EV Secretome Differentially Affects Prostate Cancer Cell Migration

**DOI:** 10.3390/cells14070547

**Published:** 2025-04-04

**Authors:** Dolma Choezom, Jan-Moritz Plum, Pradhipa Karuna M., Adi Danieli-Mackay, Christof Lenz, Phillipp Brockmeyer, Julia Christina Gross

**Affiliations:** 1Department of Hematology and Oncology, University Medical Center Goettingen, 37075 Goettingen, Germany; dolma.choezom@mpinat.mpg.de (D.C.);; 2Department of Developmental Biochemistry, University Medical Center Goettingen, 37077 Goettingen, Germany; 3Department of Clinical Chemistry, University Medical Center Goettingen, 37075 Goettingen, Germany; christof.lenz@med.uni-goettingen.de; 4Bioanalytical Mass Spectrometry, Max Planck Institute for Multidisciplinary Sciences, 37077 Goettingen, Germany; 5Department of Oral and Maxillofacial Surgery, University Medical Center Goettingen, 37075 Goettingen, Germany; 6Institute of Molecular Medicine, Department Medicine, HMU Health and Medical University Potsdam, 14471 Potsdam, Germany

**Keywords:** cell migration, extracellular matrix, neutral sphingomyelinases, *SMPD2*, SMPD3

## Abstract

Tumor-derived extracellular vesicles (EVs) play an important role in cancer progression. Neutral sphingomyelinases (nSMases) are lipid-modifying enzymes that modulate the secretion of EVs from cells. How nSMase activity and therefore ceramide generation affect the composition and functionality of secreted EVs is not fully understood. Here, we aimed to investigate the expression of nSMases 1 and 2 in prostate cancer (PCa) tissue and their role in EV composition and secretion for prostate cancer cell migration. Reduced nSMase 1 and 2 expression was found in prostate cancer and correlated with the age of the patient. When nSMase 2 was inhibited by GW4869 in PCa cells (PC3 and DU145), the EV secretome was significantly altered, while the number of EVs and the total protein content of released EVs were not significantly changed. Using proteomic analysis, we found that extracellular matrix proteins, such as SDC4 (Syndecan-4) and SRPX-2, were differentially secreted on EVs from GW4869-treated PC3 cells. In scratch wound migration assays, GW4869 significantly increased migration compared to control PC3 cells but not DU145 cells, while SDC4 knockdown significantly reduced the migration of PC3 cells. These and other nSMase-2-dependent secreted proteins are interesting candidates for understanding the role of stress-induced EVs in the progression of prostate cancer.

## 1. Introduction

Prostate cancer (PCa) is the most common malignancy and the second leading cause of cancer-related death in men. This heterogeneous disease is characterized by high variability in clinical outcomes. PCa progresses slowly and can be treated effectively when detected early. But, identifying patients likely to progress remains difficult, resulting in cases being diagnosed after the cancer has already advanced to metastasis. A better understanding of the progression of tumors and the development of approaches for earlier detection is therefore needed.

Recently, a number of studies have shown that extracellular vesicles (EVs) play important roles in PCa progression by stimulating malignant transformation, crosstalk with neighboring cells and macrophages, and osteoblast differentiation during metastasis [1]. Currently, the decision regarding non-metastasized prostate cancer treatment (curative treatments or active surveillance) is almost exclusively based on histological architecture (Gleason score) [2], prostate-specific antigen (PSA) levels, and the local disease state (TNM (tumor, nodes, metastasis)). However, early detection and routine surveillance could benefit tremendously from novel biomarkers and a better understanding of the biology of prostate cancer.

In addition to PSA, emerging biomarkers such as circulating tumor cells (CTCs), cell-free DNA (cfDNA), and non-coding RNAs (e.g., PCA3, miR-141, and miR-375) are gaining attention for their potential in prostate cancer detection and prognosis [3]. Moreover, extracellular-vesicle-associated markers such as CD9, CD63, and TSG101 have been identified as promising diagnostic tools, offering a more precise molecular signature of tumor-derived EVs [4]. These advancements could significantly improve patient stratification and treatment decisions [5].

Neutral sphingomyelinases (nSMases; gene names: *sphingomyelin (SM) phosphodiesterases, SMPDs*) hydrolyze SM to phosphocholine and ceramides (Cer). Cer function as lipid second messengers in cellular signaling pathways in tumor-suppressive and anti-proliferative processes. Ceramide synthesis by nSMase 2 is also involved in the secretion of small EVs, and thus, GW4869, an nSMase 2 inhibitor, is widely used in cell culture to inhibit exosome secretion [6]. However, the effects are cell-line-dependent, as, in breast cancer cells, GW4869 leads to an increase in larger EVs from the plasma membrane and a decrease in smaller EVs [7]. This study aims to investigate the role of nSMase2 in prostate cancer progression by analyzing its impact on EV secretion and composition. Through proteomic profiling and functional assays, we explored how nSMase2 inhibition alters the EV cargo composition and where these changes contribute to tumor cell migration. We elucidated the functional implication of the ceramide-dependent EV secretome in cancer progression. In the malignant prostate cancer cell line PC3, we found that GW4869 alters the protein content of the released EVs and supports tumor cell migration via the secretion of ECM proteins.

## 2. Materials and Methods

### 2.1. Cell Culture and Transfection

DU145 (ACC 261) and PC3 cells (ACC 465) (DSMZ, Braunschweig, Germany) were cultured under standard conditions. DU145 cells were maintained in RPMI-1640 medium (Gibco), while PC3 cells were cultured in DMEM (Gibco), both supplemented with 10% fetal calf serum (Biochrom, Berlin, Germany). Cells were incubated at 37 °C in a humidified atmosphere with 5% CO_2_. Cells were transiently transfected with Screenfect siRNA (Screenfect GmbH, Eggenstein-Leopoldshafen, Germany) according to the manufacturer’s instructions, and cells were routinely checked for mycoplasma contamination to ensure cultural integrity. Dharmacon siRNA SMARTpools (Lafayette, CO, USA) were used against the sequences in the Table 1 below.

### 2.2. Real-Time qPCR

Total RNA was extracted from cells using Trizol reagent (Sigma, St. Louis, MO, USA). A total of 5 µg of RNA was used for complementary DNA (cDNA) synthesis with the REVERT AID FIRST STRAND cDNA Synthesis Kit (NEB, Ipswich, MA, USA). The synthesized cDNA was then subjected to quantitative real-time PCR (qRT-PCR) using iQ SYBR Green SUPERMIX (Bio-Rad, Hercules, CA, USA) and specific primers targeting β-actin (Fwd: GAGCACAGAGCCTCGCCTTT, Rev: ACATGCCGGAGCCGTTGTC), SDC4 (Fwd: GGCCTTCCTCCCTTCCTTTC, Rev: CATCTCAAGGTGCAGGTGGT), and SRPX-2 (Fwd: TGGCAAAAGCGGACTATGGG, Rev: ATGGGGCCTGGTCATCTGTA). Gene expression levels were determined using the 2^−∆∆Ct^ method, with β-actin serving as the reference gene for normalization. qRT-PCR was performed on a CFX system (Bio-Rad, Hercules, CA, USA).

### 2.3. Extracellular Vesicle Purification

Extracellular vesicles were purified by differential centrifugation as described previously [4,5]. In short, conditioned-medium supernatants from mammalian cells were subjected to sequential centrifugation steps of 750× *g*, 1500× *g*, and 14,000× *g* before pelleting exosomes at 100,000× *g* in a SW41Ti swinging-bucket rotor for 2 h (Beckman, Brea, CA, USA). After discarding the supernatant, the resulting exosome pellet was dissolved in fresh PBS.

### 2.4. Immunohistochemistry

Paraffin-embedded tissue arrays (PR808, PR242b) with a total of 104 cores from 45 cases of prostatic adenocarcinoma and 7 healthy prostate tissues in duplicate with TNM, stage, Gleason score, and grade information were obtained from Biocat, Heidelberg, Germany. Arrays were heated at 50 °C to facilitate paraffin removal. Deparaffinization was carried out using a graded alcohol series, beginning with 100% xylol for 20 min, followed by 70% ethanol for 5 min. After three washes with water (5 min each), antigen retrieval was performed by heating the slides in citric acid buffer (pH 6.4) using a microwave-compatible pressure cooker. The slides were then allowed to cool, rinsed with water, and incubated in phosphate-buffered saline (PBS) containing 3% H_2_O_2_ for 45 min to inhibit endogenous peroxidase activity. Following PBS washes, the slides were blocked with bovine serum albumin (BSA) for 1 h before being incubated overnight at 4 °C with primary antibodies—SMPD2 (rabbit, Merck, Darmstadt, Germany), SMPD3 (rabbit and mouse, Santa Cruz, Dallas, TX, USA), and androgen receptor (Abcam, Cambridge, UK) as a positive control. A no-primary-antibody control was also included. Antibodies were diluted 1:50 in BSA. The next day, slides were washed with PBS-T and incubated with biotinylated secondary antibodies (1:200, ThermoFisher Scientific, Waltham, MA, USA) for 1 h at room temperature. After another PBS-T wash, they were treated with avidin-linked peroxidase (1:1000, Merck, Darmstadt, Germany) for 45 min. Immunostaining was performed using 2% 3,3′-diaminobenzidine (DAB) with 0.12‰ H_2_O_2_ in PBS, and the reaction was stopped after 3.5 min. Finally, slides were counterstained with hematoxylin, dehydrated through an alcohol series (50% xylol/isopropanol, 100% isopropanol, followed by 100%, 90%, and 70% ethanol), mounted, and left to dry overnight under a fume hood.

### 2.5. TMA Evaluation

Tissue microarray (TMA) slides were digitized using the Motic EasyScan One whole-slide scanner (Motic, Xiamen, China) at 20× magnification, achieving a resolution of 0.5 μm/pixel. Semi-automated immunohistochemical analysis was performed using QuPath software (v0.2.0-m8) [6]. Tissue cores were identified through an automated TMA dearrayer, with manual verification and adjustments as necessary. Stain separation was optimized using color deconvolution, incorporating spot vector and background estimates for accuracy [7]. Automated cell and membrane detection was employed to identify cells within TMA cores, where cell boundaries were estimated based on constrained nuclear expansion. A total of 33 intensity and morphological parameters were measured, including nuclear area, circularity, staining intensities for hematoxylin and DAB, and the nucleus-to-cell area ratio. From these, 16 key features were empirically selected for further analysis, complemented by local cell density metrics using Gaussian-weighted smoothing. To differentiate tumor cells from other cell types (e.g., stromal cells), a two-way random tree classifier was trained. Tumor cells were further classified into negative, weakly positive, moderately positive, or strongly positive based on the mean DAB optical density. H-scores (ranging from 0 to 300) were calculated as follows: (3× % strongly stained) + (2× % moderately stained) + (1× % weakly stained) [8]. Biomarker expression (H-score values) was analyzed in relation to clinical characteristics, including age, TNM stage, histopathological grading, UICC stage, and Gleason score, using Mann–Whitney tests. Age groups were classified as <70 or ≥70 years. T2, T3, and T4 tumor stages were consolidated into two categories, and UICC stages II and III–IV were combined. Lymph node involvement (N) and distant metastasis (M) were categorized as either positive or negative. Gleason scores were dichotomized as ≤6 or >6. Statistical analyses were conducted using Prism software (version 10.4.1, GraphPad, Boston, MA, USA), with a significance level of α = 5%.

### 2.6. Immunoblot

To analyze total cell lysates via immunoblotting, cells were first lysed to extract proteins. The lysates were then mixed with SDS-PAGE sample buffer and heated at 95 °C for 5 min to denature proteins and ensure uniform charge distribution. Proteins were separated on 4–12% gradient gels (Bolt Bis-Tris Plus Gels, Thermo Scientific, Waltham, MA, USA) and subsequently transferred onto PVDF membranes (Merck, Darmstadt, Germany). Following blocking with 5% (wt/vol) milk in TBST, membranes were incubated overnight with primary antibodies. The next day, they were washed with TBST and incubated for 1 h with secondary antibodies conjugated to LICOR 680 nm or 800 nm fluorophores. After final washes in TBST, signal detection was performed using a LICOR imaging system (Lincoln, NE, USA).

### 2.7. Nanoparticle Tracking Analysis

Extracellular vesicle (EV) samples were diluted 1:25 in PBS by mixing 16 μL of the sample with 384 μL of PBS. The diluted solution was then injected into the measurement chamber of the NanoSight LM10 system (Malvern Instruments, Malvern, UK). Video recordings of 60 s were captured using the NanoSight NTA 2.3 Analytical Software and an sCMOS camera (Oxford Instruments, Abingdon, UK), with the following settings: shutter speed 30.01 ms, gain 500, frame rate 24.99 fps, and temperature 22.5 °C. For each sample, three separate videos were recorded, ensuring at least 200 detectable particle tracks per video. The measurement position was slightly adjusted between recordings to account for sample heterogeneity. The recorded videos were analyzed using NanoSight NTA 2.3 Analytical Software, providing data on particle concentration, size distribution, mean size, and mode size of the EV samples.

### 2.8. Viability Assay

Cell viability after GW4869 treatment was measured by performing CellTiter-Glo assays (Promega, Madison, WI, USA): Cells were seeded in a 96-well plate. After treatment with GW4869 or DMSO for 16 h, 100 μL of the cell titer glow reagent was added to each well (1:1). The plate was incubated on a shaker for 2 min at RT to allow cell lysis and then incubated without shaking for 10 min to allow luminescence signal stabilization. The signal was measured using a luminometer, and data were analyzed using MikroWin 2000 lite version 4.43.

### 2.9. Proliferation and Migration Assay

PC3 and DU145 cells were seeded into 96-well plates at varying densities, ranging from 5 k to 10 k, 20 k, 40 k, and 50 k cells/well, and then incubated for 24 h at 37.5 °C and 5% CO_2_ to allow cell attachment. A wound of defined width was created using the WoundMaker tool (IncuCyte, Sartorius, Goettingen, Germany), and cells were subsequently washed with fresh medium (DMEM for PC3, RPMI1640 for DU145) to remove cell debris. Cells were then treated with DMSO (vehicle control) or 10 µM GW4869 or untreated medium. Plates were then placed in the IncuCyte S3 live cell imaging system at 37 °C and 5% CO_2_, and images of every well were taken every hour for 24 h to track wound closure. The rate of migration was analyzed by measuring the reduction in wound area over time using IncuCyte software (v2018B).

### 2.10. Mass Spectrometry

Samples were reconstituted in 1× NuPAGE LDS Sample Buffer (Invitrogen, Waltham, MA, USA) and applied to 4–12% NuPAGE Novex Bis-Tris Minigels (Invitrogen). Samples were run 1 cm into the gel for purification and stained with Coomassie Blue for visualization purposes. After washing, gel slices were reduced with dithiothreitol (DTT), alkylated with 2-iodoacetamide, and digested with trypsin overnight. The resulting peptide mixtures were then extracted, dried in a SpeedVac (Eppendorf, Hamburg, Germany) reconstituted in 2% acetonitrile/0.1% formic acid (*v*:*v*), and prepared for nanoLC-MS/MS as described previously [9]. All samples were spiked with a synthetic peptide standard used for retention time alignment (iRT Standard, Schlieren, Switzerland). Protein digests were analyzed on a nanoflow chromatography system (Eksigent nanoLC425, Sciex, Framingham, MA, USA) hyphenated to a hybrid triple-quadrupole–TOF mass spectrometer (TripleTOF 5600+, Sciex) equipped with a Nanospray III ion source (Ionspray Voltage 2400 V, Interface Heater Temperature 150 °C, Sheath Gas Setting 12) and controlled by Analyst TF 1.7.1 software build 1163 (all Sciex). In brief, peptides were dissolved in loading buffer (2% acetonitrile, 0.1% formic acid in water) to a concentration of 0.3 µg/µL. For each analysis, 1.5 µg of digested protein was enriched on a precolumn (0.18 mm ID × 20 mm, Symmetry C18, 5 µm, Waters, Milford, MA, USA) and separated on an analytical RP-C18 column (0.075 mm ID × 250 mm, HSS T3, 1.8 µm, Waters) using a 90 min linear gradient of 5–35% acetonitrile/0.1% formic acid (*v*:*v*) at 300 nL min^−1^.

Qualitative LC/MS/MS analysis was performed using a Top25 data-dependent acquisition method with an MS survey scan of *m*/*z* 350–1250 accumulated for 350 ms at a resolution of 30,000 full width at half maximum (FWHM). MS/MS scans of *m*/*z* 180–1600 were accumulated for 100 ms at a resolution of 17,500 FWHM and a precursor isolation width of 0.7 FWHM, resulting in a total cycle time of 2.9 s. Precursors above a threshold MS intensity of 125 cps with charge states 2+, 3+, and 4+ were selected for MS/MS, and the dynamic exclusion time was set to 30 s. MS/MS activation was achieved by CID using nitrogen as a collision gas and the manufacturer’s default rolling collision energy settings. Two technical replicates per sample were analyzed to construct a spectral library.

For quantitative SWATH analysis, MS/MS data were acquired using 65 variable-size windows [10] across the 400–1050 *m*/*z* range. Fragments were produced using rolling collision energy settings for charge state 2+, and fragments were acquired over an *m*/*z* range of 350–1400 for 40 ms per segment. Including a 100 ms survey scan, this resulted in an overall cycle time of 2.75 s. For each biological state, 3 × 3 replicates were acquired.

Protein identification was achieved using ProteinPilot Software version 5.0 build 4769 (Sciex) with “thorough” settings. A total of 322,587 MS/MS spectra from the combined qualitative analyses were searched against the UniProtKB human reference proteome (revision 02-2017, 92,928 entries) augmented with a set of 52 known common laboratory contaminants to identify 1651 proteins at a false discovery rate (FDR) of 1%.

Spectral library generation and SWATH peak extraction were achieved in PeakView Software version 2.1 build 11041 (Sciex) using the SWATH quantitation microApp version 2.0 build 2003. Following retention time correction using the iRT standard, peak areas were extracted using information from the MS/MS library at an FDR of 1% [11]. The resulting peak areas were then summed to peptide and, finally, 1495 protein area values per injection, which were used for further statistical analysis.

### 2.11. Bioinformatics Analysis

Heatmaps were generated with Morpheus (https://software.broadinstitute.org/morpheus (accessed on 26 January 2021)). Go term and network analysis was performed with the String database [12], and ID conversion was performed with David [13].

### 2.12. Statistics

All experiments were carried out at least in biological triplicates. Error bars indicate standard deviations. Statistical significance was calculated by carrying out One-way ANOVA or Student’s *t*-test where appropriate. The graphs were generated and statistical analyses were performed using GraphPad Prism 6. The mass spectrometry data that support the findings of this study will be available at the public repository PRIDE.

## 3. Results

### 3.1. Reduced SMPD2 and SMPD3 Expression in Prostate Cancer

To investigate the role of nSMase 1 and nSMase 2 (gene names: *SMPD2* and *SMPD3*) in prostate cancer progression, we evaluated their protein levels via immunohistochemical staining of prostate cancer tissues and prostate tissue arrays (n = 104) (Figure 1A,B), with the androgen receptor (AR) for positive control staining (Appendix A). Antibody staining was evaluated semi-automatically using quPath [14], and the intensity correlated with age, TNM parameters, histopathological grading, UICC stage, and Gleason score based on the H-score values (0–300). In men over 70 years, we found significantly lower SMPD2, SMPD3, and AR levels than in men under 70 (Figure 1C–F), consistent with declining AR expression in prostate tissue with age [2].

In addition, we analyzed the expression of *SMPD2* and *SMPD3* in prostate and prostate cancer samples from the proteinatlas database [15]. Similarly to the immunohistochemical staining of *SMPD2* and *SMPD3*, lower expression of SMPD2 or SMPD3 was observed in prostate cancer versus healthy tissue (Figure 2A,B).

To establish a pipeline for biomarker identification for prostate cancer, we previously compared EVs secreted from malignant PC3 and DU145 cells, from bone and brain metastases, respectively, to EVs from benign prostate PTNA1 cells in a proteomics screen [16]. When comparing *SMPD2* and *SMPD3* levels in seven prostate cancer cell lines from proteinatlas data [15], DU145 and PC3 were in the middle-to-low range of expression, and *SMPD2* levels were higher than *SMPD3* levels (Figure 2C). By Western blot analysis, we confirmed that the protein levels of *SMPD2* and *SMPD3* in DU145 and PC3 cell lysates were detectable (Figure 2D,E).

We next tested the functional effect of nSMase 2 on cancer cell migration by using GW4869, an inhibitor of nSMase 2 activity [6]. Cell viability under 5 µM GW4869 was unchanged in DMSO-treated control cells after 24 h (Appendix A). In a scratch wound assay (Figure 2F–I), the migration speed differed between the two cell lines. PC3 cells reached 40% wound closure after 24 h, whereas DU145 wounds were closed after 24 to 36 h. Interestingly, in the presence of GW4869, wound closure after 12 to 24 h was significantly faster in PC3 cells than in DMSO-treated control cells (Figure 2F,G). In contrast, DU145 cells’ wound closure was independent of GW4869 treatment (Figure 2H,I). Taken together, we found lower levels of nSMases in cancer tissue and samples from older patients and found that the reduced activity of nSMase 2 contributed to the malignant behavior of PC3 cells but not DU145 cells.

### 3.2. Ceramide-Dependent EV Secretion from PC3 Cells

Extracellular vesicles (EVs) are small membrane particles of 50–450 nm, secreted from different cell types. Recently, a number of studies have shown that EVs from malignant cells and their protein composition stimulate cell migration and invasion [17]. Thus, we next focused on the role of nSMase 2 activity in EV secretion from prostate cancer cells. From the supernatants of PC3 and DU145 cells treated with GW4869 or DMSO in exo-free DMEM for 24 h, we purified EVs by differential centrifugation with 14,000× *g* and 100,000× *g* steps to purify large and small EVs [18]. Nanoparticle tracking analysis revealed overlapping size profiles of large/P14-EVs (130–350 nm) and small/P100-EVs (50–210 nm) for both cell lines (Figure 3A–E, Appendix A–G).

Inhibition of nSMase 2 activity supposedly blocks EV secretion, but recently, we found that GW4869 differentially affects the amount of EVs secreted [7]. For PC3 cells, the concentration of larger P14-EVs was unchanged by GW4869 treatment, while P100-EVs were slightly increased in the size range of 110–150 nm (Figure 3C,E). The secretion of P14 and P100-EVs from DU145 was not changed by GW4869 (Appendix A–G). Western blot analysis of the cell lysate and P100 showed that Calnexin did not contaminate the P100 fractions in either cell line. Alix, CD63, and Syntenin, all markers for small EVs, were enriched in EVs from GW4869-treated cells, while CD81 was reduced in P100 (Figure 3F–H). Thus, rather than blocking EV secretion, inhibition of nSMase 2 activity might alter the protein composition of secreted EVs from PCa cells.

### 3.3. Ceramide-Dependent EV Proteome

Next, we subjected EV preparations from DMSO- and GW4869-treated PC3 and DU145 cells to label-free mass spectrometry-based proteomics, with subsequent quality control and candidate selection (Figure 4A). We first investigated the overall composition of EVs secreted from both prostate cell lines in the two purified EV fractions, P14 and P100. A full list of all identified proteins is given in Appendix A. Upon the analysis of all 1495 proteins quantified, with a false discovery rate of 1% for both EV fractions and both replicates from DMSO-treated cells, we identified 172 proteins on P14-EVs and 217 on P100-EVs from PC3, of which 137 overlapped. From the 137 overlapping proteins in PC3-EVs, 60 were known TOP EV proteins (Figure 4B). From DU145 cells, 252 proteins were identified in P14-EVs and 221 in P100-EVs, of which 178 overlapped, with 47 being TOP EV proteins (http://microvesicles.org/extracellular_vesicle_markers (accessed on 26 January 2021)) (Figure 4C). This analysis is a proof of principle for our EV isolation technique and shows that, in both P14 and P100 fractions, the abundant proteins from Mass Spec overlap with the Top EV markers. In addition, our method is in line with EV isolation guidelines [19].

Our next aim was to see which EV markers were enriched in the GW4869 treatment compared to DMSO. This involved calculating differentially regulated markers that were 1.5-fold enriched upon GW4869 treatment. From PC3 cells, a total of 49 proteins were enriched in P14 and 93 proteins in the P100 fraction with GW4869, of which 33 overlapped (Figure 4D).

From DU145 cells, 171 proteins were enriched in the P14 fraction and 36 enriched in the P100 fraction with GW4869, with 26 proteins overlapping (Figure 4E). Interestingly, of these 33 and 26 overlapping proteins between P14 and P100 EVs from each PC3 and DU145 cell line upon GW4869 treatment, 13 were found to be upregulated in both EV fractions from both cell lines. As both P14 EV and P100 EV could contribute to tumor cell migration, the overlap between them was taken for a further analysis of factors. String database annotation showed a strong connection between the identified proteins (Figure 4G,H). Gene ontology (GO) term and Reactome database analyses revealed a >20-fold enrichment of cellular components, such as extracellular matrix components, collagens, and laminins, and glycan degradation in the GW4869-P100 and P14 fractions in both cell lines (Figure 5A–D). Thus, upon nSMase 2 inhibition, ECM components were secreted in EVs from both PCa cell lines.

### 3.4. Ceramide-Dependent Tumor Migration

The dependence on the purification technique and the biological similarity of P14- and P100-EVs has led to current EV research treating these not as individual populations but as a heterogeneous mixture of EVs [19]. Indeed, from the significantly quantified proteins in both cell lines and both fractions, we found 20 proteins enriched in PC3 EV fractions, 13 proteins in both fractions in both cell lines, and 13 exclusively upregulated in DU145 EV fractions (Figure 6A). Among the components differentially affected by nSMase 2 inhibition were several laminins, glypican, and other matrix proteins (Appendix A). Finally, we selected two interesting proteins enriched in PC3 EVs in a ceramide-dependent manner, SDC4 and SRPX2, as candidates to further characterize in the wound-healing assay. SRPX2, a ligand for the urokinase plasminogen activator surface receptor [20], is commonly overexpressed in several types of cancer, including prostate cancer [19,20,21]. Interestingly, SRPX2 promotes cell migration and adhesion through focal adhesion kinase (FAK) phosphorylation [18,19]. Similarly, SDC4 is a proteoglycan involved in the generation of sEVs from endosomal compartments [22,23], and its high expression is involved in cancer cell invasion [24,25]. To elucidate whether the two candidates contributed to cell migration by being sorted into EVs in an nSMase-2-dependent manner, we created phenocopies by siRNA-mediated knockdown (KD) in both prostate cancer cells. SDC4 and SRPX2 siRNAs were transiently transfected into DU145 and PC3 cells, and knockdown efficiency was more than 50%, as shown by quantitative PCR (Figure 6B,C,F,G). While SRPX2 KD had no effect on PCa cell migration, SDC4 KD led to significantly slower wound closure in both cell lines, with a stronger effect in PC3 cells (Figure 6D,E,H,I). Taken together, our results suggest an nSMase-2-dependent role for EV-sorted matrix components that might be functionally relevant in prostate cancer cell migration.

## 4. Discussion

Our study provides new insights into the role of neutral sphingomyelinases (nSMases) (*SMPD2* and *SMPD3*) in prostate cancer progression and extracellular vesicle (EV) secretion. We found that the reduced expression of nSMases in prostate cancer correlates with age. NSMase2 inhibition alters the composition of EVs secreted from prostate cancer cells. In particular, extracellular matrix proteins, including Syndecan-4 (SDC4) and SRPX-2, are differentially secreted in EVs, as our proteomic analysis revealed. NSMase2 inhibition enhanced the migration of PC3 but not DU145 cells, which was reduced in the SDC4 KD experiment. The results demonstrate that *SMPD2* and *SMPD3* expression is reduced in prostate cancer, indicating that a potential disruption in sphingomyelin metabolism and EV secretion may contribute to tumor progression.

### 4.1. nSMase Suppression and Prostate Cancer Progression

Our first significant finding was the reduced expression of *SMPD2* and *SMPD3* in prostate cancer tissues and samples from older patients. This downregulation suggests a possible tumor-suppressive role of these enzymes, aligning with previous studies that have implicated sphingolipid metabolism in stress pathways and apoptosis induction [26]. Specifically, nSMase 2 is considered a major candidate for mediating the stress-induced production of ceramide, leading to apoptosis [27]. The inhibition of nSMase 2 activity using GW4869 in PC3 and DU145 cells resulted in an altered EV composition; similarly, we previously reported that GW4869 changes EV secretion from breast cancer cells, resulting in the changed metabolic and proteomic profiles of these EVs [7]. This is further supported by the hypothesis that nSMases play a critical role in modulating EV cargo release, as shown by recent proteomic studies [17]. In a recent meta-study of lipid profiling of patient samples of different tumor entities, it was demonstrated that low values in tissue samples and high values in liquid samples indicated a poor prognostic trend, a fact that is in agreement with cancer cells secreting EVs, which contain a characteristic set of proteins and lipids [28]. Our results are in line with the idea that altered EV secretion from cancer cells influences the tumor microenvironment, shifting it toward tumor progression, and hold promise for a better understanding of cancer biology as well as disease monitoring.

### 4.2. EV Secretion and Cell Migration

Upon nSMase2 inhibition, altered EV secretion coincided with an increase in the migration of PC3 cells, a hallmark of cancer metastasis. This finding is particularly notable, as EVs are known to carry signaling molecules that can modulate the behavior of recipient cells, including metabolism and invasion [29]. The enhanced migration seen in our study suggests that nSMase 2 inhibition may enhance the pro-tumorigenic potential of prostate cancer cells by altering the composition of secreted EVs. Signaling molecules on EVs that mediate migration are cell adhesion molecules and ECM proteins (such as integrins and fibronectin), which promote cell persistence and efficient directional movement [30]. The specific mechanisms by which the here-identified proteins regulate cell migration, however, warrant further investigation.

### 4.3. NSMase-Dependent EV Proteome: Cell Type Specificity

Our proteomic analysis of EVs revealed an nSMase-2-dependent, cell-type-specific EV proteome. This finding is important, as it suggests that the role of *SMPD2* and *SMPD3* in EV biogenesis and protein cargo loading may vary between different prostate cancer cell lines. Proteins enriched in EVs, including those involved in extracellular matrix (ECM) remodeling and cell adhesion (e.g., laminins and glypicans), could be key drivers of tumor cell behavior, facilitating processes such as cell migration and invasion. In a previous study, we compared EVs secreted from malignant PC3 cells and benign PNT1A cells and found 64 proteins exclusively in PC3-derived sEVs, including claudin 3, which enabled the discrimination of plasma from prostate cancer patients from healthy donors [16]. The enrichment of these ECM proteins within EVs supports the idea that nSMase 2 activity is modulated upon tumor progression and may regulate the tumor microenvironment through EV-mediated ECM remodeling. Identifying factors that are detectable in the plasma and discriminate between early and late or local and metastasizing tumors may help to assess and guide decisions for prostate cancer treatment [31].

### 4.4. Functional Roles of SDC4 and SRPX2

Two proteins identified in the EV proteome, SDC4 and SRPX2, were further characterized due to their potential relevance to cancer progression, as seen in other cancer entities, such as thyroid, pancreas, and gastric cancer [18,19,24,25,32]. The knockdown of SDC4 in PC3 cells significantly reduced cell migration, as demonstrated by wound-healing assays. This suggests that SDC4 plays a critical role in regulating cell motility, potentially through its involvement in ECM interactions and cell adhesion processes [33]. SRPX2 has been previously implicated in prostate cancer [21], and its presence in EVs may suggest a role in promoting metastasis [20]. Syndecan and syntenin expression at various stages of prostate cancer varies according to the genetic heterogeneity of the tumors [34]. These and other nSMase-2-dependent secreted proteins are interesting candidates for understanding the role of stress-induced EVs in the progression of prostate cancer and might have biomarker potential if they are secreted at an early stage of cancer progression. Further studies are needed to elucidate the exact function of EV-secreted proteins in prostate cancer.

### 4.5. Conclusions

Our findings highlight the importance of nSMase 2 in modulating EV secretion and proteome composition in prostate cancer. Similar to the cell-type-specific regulation of lipid metabolism, the cell-type-specific EV proteome suggests that different prostate cancer cell types may rely on distinct EV-mediated mechanisms for progression and metastasis. This opens up the possibility of targeting nSMase 2 pathways or EV-associated proteins like SDC4 as potential therapeutic strategies for limiting prostate cancer metastasis. A recent study showed that using cell-line-derived EV-based proteins for PCa diagnostics can help to identify markers that discriminate metastasizing from local PCa [35].

Future research should focus on the mechanistic details of how nSMase 2 regulates EV cargo selection and secretion and how these EVs influence the tumor microenvironment. Additionally, the targeting of SDC4 and SRPX2 in prostate cancer should be explored further, particularly in relation to their roles in cell migration and metastasis. While our findings provide novel insights into nSMase2-dependent EV secretion and its role in prostate cancer migration, this study was limited to in vitro models. Further studies should validate these findings in vivo to confirm the physiological relevance of our observations. Additionally, the impact of nSMase2 inhibition on EV secretion and tumor progression in a complex tumor microenvironment remains to be explored.

## Figures and Tables

**Figure 1 cells-14-00547-f001:**
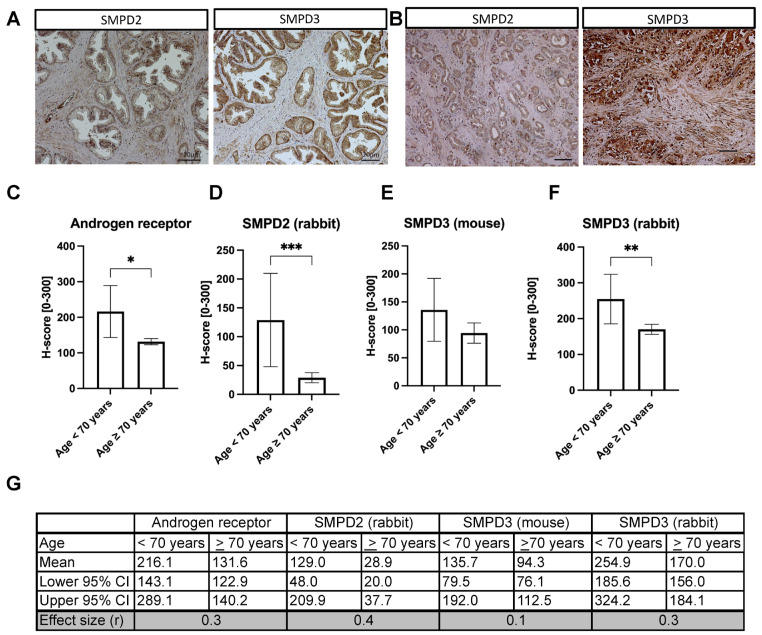
*SMPD2* and *SMPD3* expression in prostate cancer. (**A**) Paraffin-embedded tissue arrays (52 duplicate cores) were stained for SMPD2 and SMPD3 (mouse antibodies) in healthy prostate tissue (n = 7) and (**B**) prostatic adenocarcinoma tissue (n = 45), scale bars = 10 µm; (**C**) androgen receptor, (**D**) SMPD2, and (**E**,**F**) SMPD3 stratified by age (<70 and ≥70 years). Median SMPD3, SMPD2, and androgen receptor levels significantly vary between age groups above and below 70 years; Mann–Whitney, statistical significance * *p* < 0.05, ** *p* < 0.01, *** *p* < 0.001 as indicated. (**G**) The corresponding means, 95% confidence intervals, and effect sizes from (**C**–**F**).

**Figure 2 cells-14-00547-f002:**
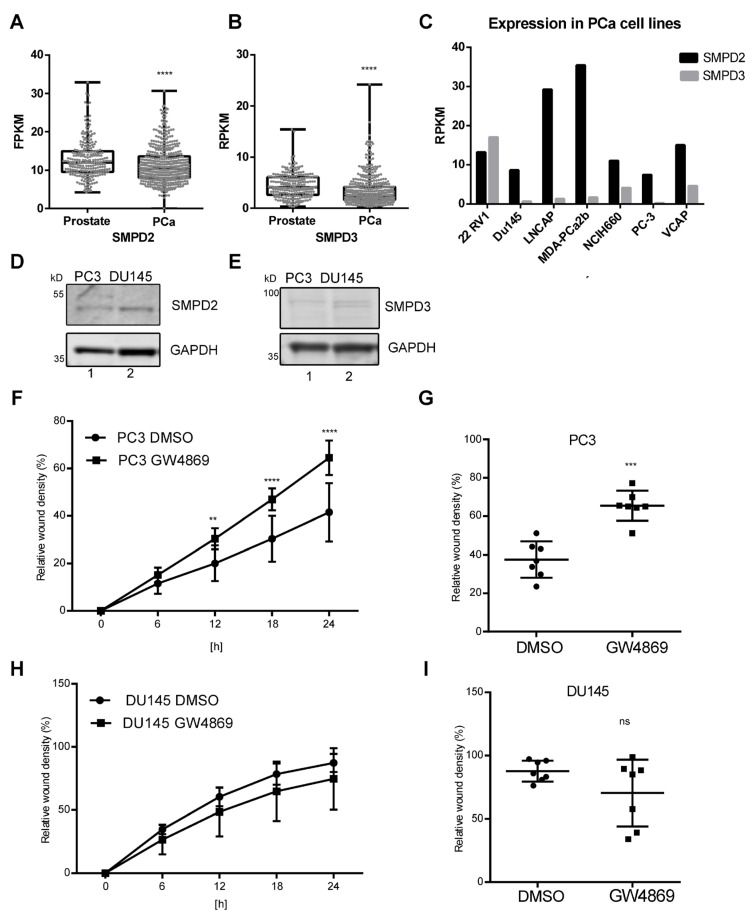
*SMPD2* and *SMPD3* expression in prostate cancer tissue and cell lines. (**A**) Data on *SMPD2* and (**B**) *SMPD3* expression in patient samples of prostate tissue (n = 245) and prostate cancer (n = 494) in fragments per kilobase per million mapped fragments from proteinatlas.org. Unpaired *t*-test with Welch’s correction, *p* < 0.0001. (**C**) Data on *SMPD2* and *SMPD3* expression in prostate cancer cell lines from proteinatlas.org. (**D**) Protein levels of SMDP2 and (**E**) *SMPD3* in PC3 and DU145 cells; GAPDH was used as a loading control. (**F**–**I**) Scratch-wound-healing assays in (**F**) PC3 cells or (**H**) DU145 cells treated with DMSO or 5 µM GW4869. The relative wound density was measured in percent. (**G**) Single data points from independent scratch wounds in PC3 or (**I**) DU145 cells after 24 h. Paired *t*-test, significance level: ns not significant, ** *p*< 0.01, *** *p* < 0.001, **** *p* < 0.0001.

**Figure 3 cells-14-00547-f003:**
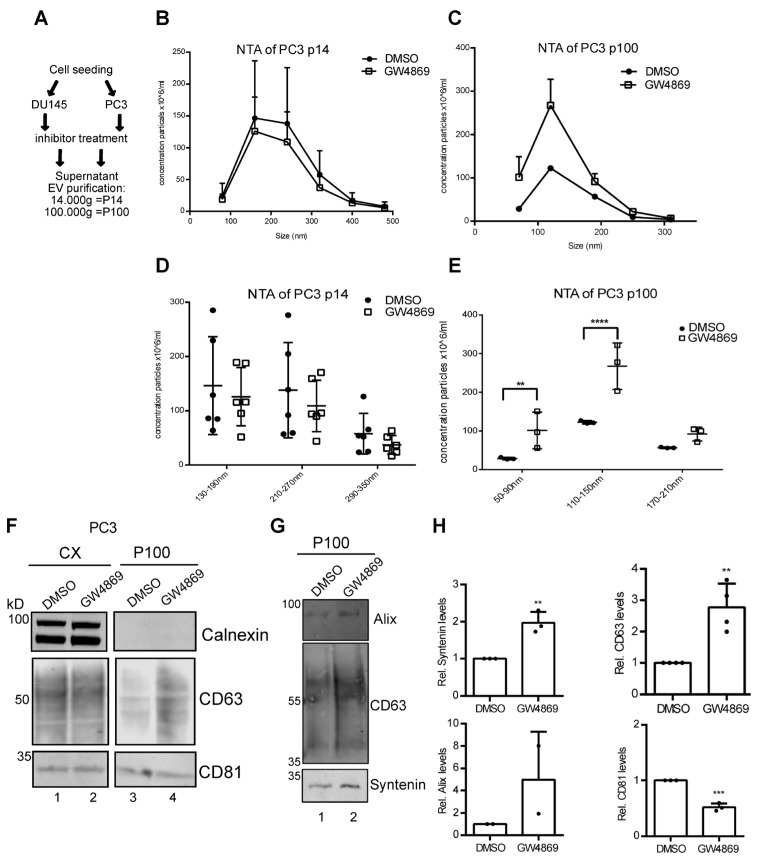
NSMase-2-dependent EV secretion from PC3 cells. (**A**) Experimental setup of EV purification (**B**) A comparison of the size distribution of secreted particles in P14 and (**C**) P100 after the treatment of PC3 cells with DMSO or GW4869, with particle diameter in nm and particle concentration in ×10⁶/mL. (**D**) Concentration differences in P14 with three size bins, 130–350 nm, and (**E**) P100 with three size bins, 50–210 nm, and their multiples compared to the control group. Statistical analysis was performed using a 2-way ANOVA with multiple testing. (**F**,**G**) Western blots of lysate and p100 of PC3 cells with Calnexin, CD63, CD81, Alix, and Syntenin. (**H**) Quantification of (**F**,**G**). Student’s *t*-test, significance level: ns not significant, ** *p*< 0.05, *** *p* < 0.001, **** *p* < 0.0001.

**Figure 4 cells-14-00547-f004:**
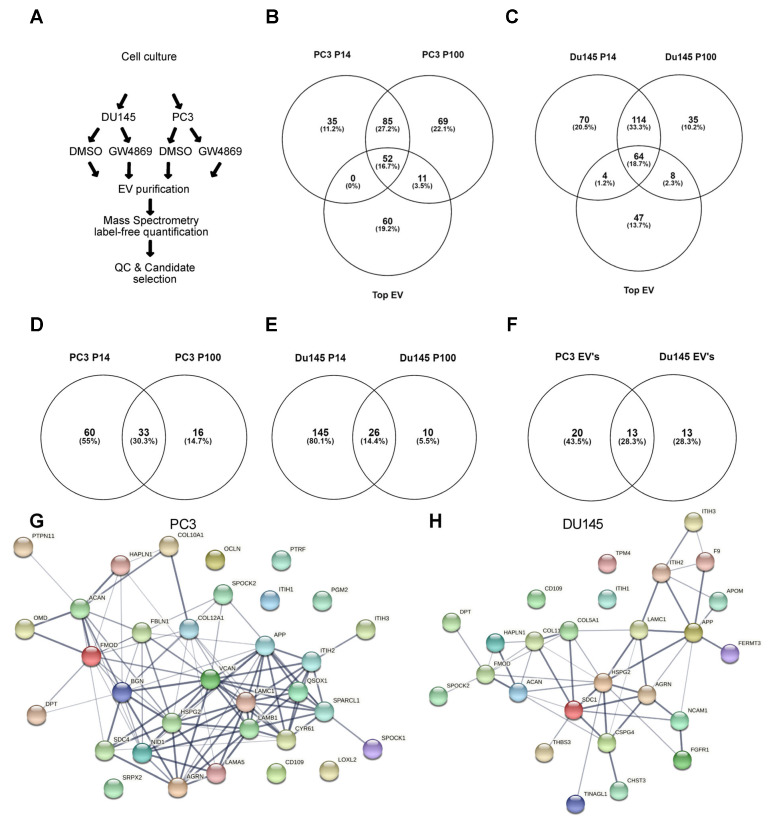
Differential EV proteomes from PC3 and DU145 cells were investigated by label-free mass spectrometry-based proteomics. (**A**) Experimental procedure with subsequent quality control and candidate selection. (**B**) Overlap of identified P14 and P100 proteins with TOP EV proteins from PC3 cells and (**C**) from DU145 cells. (**D**) Overlap of proteins enriched upon GW4869 treatment in P14 and P100 fractions in PC3 and (**E**) DU145 cells. (**F**) Of the 33 overlapping PC3 EV proteins, 13 overlapped with 26 DU145 EV proteins. (**G**) String database analysis of these 33 proteins from PC3 and (**H**) 26 proteins identified on EV from DU145 cells.

**Figure 5 cells-14-00547-f005:**
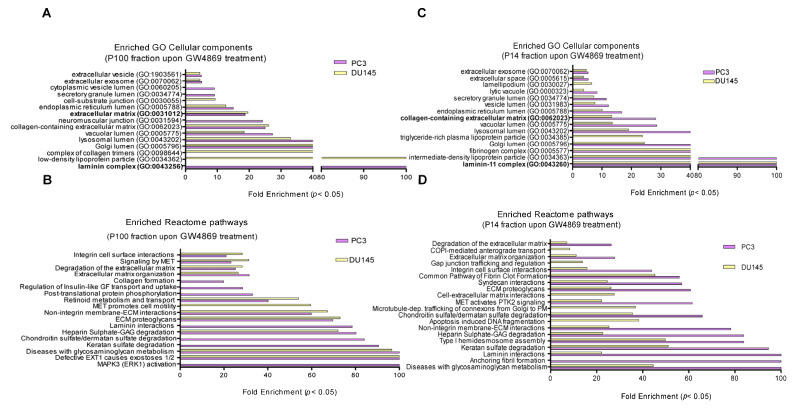
NSMase-2-dependent enrichment of cellular components and pathways. GO-term cellular components and Reactome pathway enrichment of proteins upregulated in GW4869-treated cells. (**A**) GO-term cellular components and (**B**) Reactome pathways in P100-EVs from PC3 and DU145 cells. (**C**) GO-term cellular components and (**D**) Reactome pathways in P14-EVs from PC3 and DU145 cells.

**Figure 6 cells-14-00547-f006:**
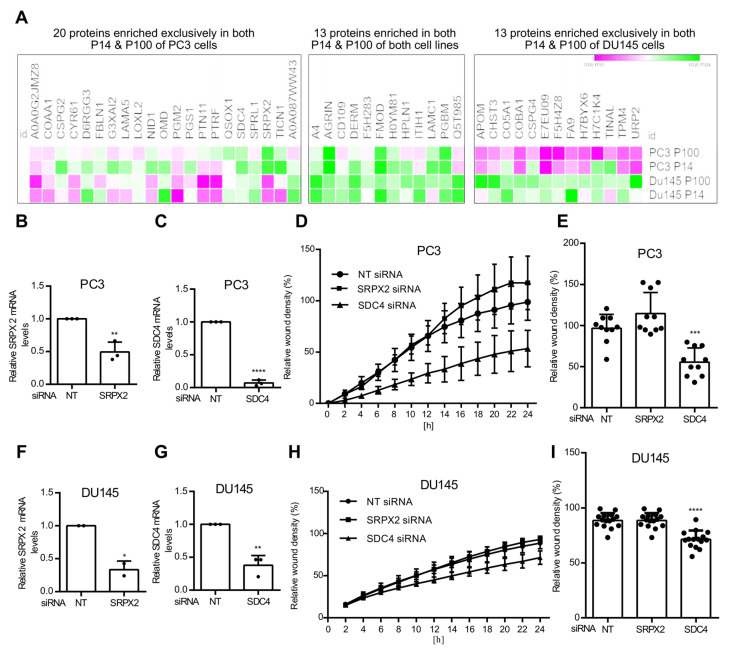
SDC4 knockdown affects PCa cell migration. (**A**) Heatmap of nSMase-2-dependent differentially regulated proteins in P14 and P100 from PC3 and DU145 cells. (**B**) Scratch-wound-healing assays of PC3 cells against SRPX2 or (**C**) SDC4 were treated with siRNA for 24 h before scratching Relative knockdown efficiency was determined by quantitative PCR. (**D**) Relative wound density was measured in percent after 24 h in PC3. (**E**) Single data points from independent scratch wounds in PC3 after 24 h. Similarly, knockdown efficiency was determined in DU145 cells (**F**) against SRPX2 or (**G**) SDC4, and then (**I**) relative wound density was measured in percent after 24 h. (**J**) Single data points from independent scratch wounds show a significant decrease in wound density for SDC4 after 24 h. Significance level: ns not significant, * *p*< 0.05, ** *p*< 0.01, *** *p* < 0.001, **** *p* < 0.0001.

**Table 1 cells-14-00547-t001:** Gene symbol, ID and Sequnces of Dharmacon siRNA SMARTpools.

Gene Symbol	Gene ID	Sequence
*SDC4*	6385	UAGAGGAGAAUGAGGUUAU GAUCGGCCCUGAAGUUGUC CCAACAAGGUGUCAAUGUC GUGAGGAUGUGUCCAACAA
*SRPX2*	27286	GUUGUGAGCUCUCCUGUGA UGAAAGCUACAAUGAAGUA CCUAUGAAGAUUAACGUCA GAUGAGAUGCCACGCACUA

## Data Availability

The mass spectrometry proteomics data have been deposited to the ProteomeXchange Consortium via the PRIDE [1] partner repository with the dataset identifier PXD062469.

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
