# Peer review of "The Ceramide-Dependent EV Secretome Differentially Affects Prostate Cancer Cell Migration"

_cells, 2025, doi:10.3390/cells14070547_

Round 1

Reviewer 1 Report

Comments and Suggestions for Authors

The manuscript should be revised as per the below comments: 

1. A clearer explanation of how the reported changes in nSMase expression and EV secretion may be converted into medicinal purposes or prostate cancer indicators should be included in the text.
2. Although the study identifies alterations in EV composition and cell migration, the paper might benefit from a more thorough investigation of the molecular pathways that connect nSMase inhibition to these results.
3. To further clarify the results and rule out non-specific effects of GW4869, the study might benefit from adding more controls, such as non-cancerous prostate cells.
4. The paper should note any possible drawbacks, such as its dependence upon vitro models and the requirement for confirmation using in vivo models or patient-derived material.
5. To strengthen the arguments made from the data, provide more thorough statistical analysis that includes effect sizes and confidence ranges.
6. To assist readers quickly grasp the main conclusions, certain graphics, especially those displaying proteomic data, might use improved labeling and annotations. Additionally, enhance the images' visual quality because some of their text is unreadable.

Reviewer 2 Report

Comments and Suggestions for Authors

hello

thank you for an interesting paper

the title is sound, clear aim is set

abstract is sufficient, nothing to add

key words used are ok, however I would add two more words - to make it 5

at the end of the abstract write the aim of the study

title and used key words and references are sufficient

abstract doesn't need correction

title is quite alright

-chapter 1

introduction is nicely prepared - its short

authors should add some more factors about prostate cancer

also an information of possible markers and diagnostics should be add

used references and structure of introduction is short, however used references are quite good

at the end of the paper there is no clear aim of the paper

please add the paper aim, or hypothesis if such is possible to write

introduction needs changes

-chapter-2-

I'm missing study inclusion and exclusion criteria

I'm missing any information about the samples, where to they came from?

how and when did authors gather those samples?

material and methods are not that clear, they have some 

methodology is somehow hard to understand - please re-write it a step by step aspect

does all sam of prostate cancer were studied by authors, or perhaps some other factors are deciding on the sample evaluation?

-results-3-

are quite alright

used figure are very well designed

text is well arranged and defined

figure 3-6 are well designed and described

results seems to be accurate

what statistical software was used for statistical analysis?

-discussion-4-

its very short

doesn't discus both authors and other paper researches

authors don't present top 5 key highlighted most important results

how authors results impact on prostate cancer and was it similar to other worlds studies?

authos didn't present any study limitations - please add them

discussion is not comparing, nor discussing or evaluating other author studies

should be re-written

-final conclusions- are ok

used references are OK

paper is very interesting

but it needs more improvement to show its scientific value

thank you very much

Round 2

Reviewer 2 Report

Comments and Suggestions for Authors

thank you

all necessary changes are made

no comments needed

Comments on the Quality of English Language

thank you

all necessary changes are made

no comments needed